# The Effect of Non-Overlapping Somatic Mutations in *BRAF*, *NRAS*, *NF1*, or *CKIT* on the Incidence and Outcome of Brain Metastases during Immune Checkpoint Inhibitor Therapy of Metastatic Melanoma

**DOI:** 10.3390/cancers16030594

**Published:** 2024-01-30

**Authors:** Wolfram Samlowski

**Affiliations:** 1Comprehensive Cancer Centers of Nevada, Las Vegas, NV 89148, USA; wsamlowski1@gmail.com; Tel.: +1-702-321-3930; 2Kirk Kerkorian School of Medicine, University of Nevada Las Vegas (UNLV), Las Vegas, NV 89106, USA; 3School of Medicine, University of Nevada (Reno), Reno, NV 89557, USA

**Keywords:** central nervous system, ipilimumab, nivolumab, pembrolizumab, “driver” mutation

## Abstract

**Simple Summary:**

Patients with metastatic melanoma have a high risk of brain involvement (brain metastases). It is not currently known whether the genetic subtype of melanoma affects the risk of brain metastases. We evaluated 85 patients who had mutation testing of their melanoma. All patients were treated with immunotherapy. Only 20% developed brain metastases. Patients with *BRAF* and *NF1* mutations were the only ones who had brain metastases when metastatic melanoma was diagnosed (12.9%). Rare patients with *BRAF*, *NRAS*, *NF1*, or *CKIT* mutations developed delayed brain metastases following immunotherapy (7.1%). Patients who did not have any of these four mutations did not develop brain metastases. Patients who developed brain metastases from melanoma had a lower survival rate than those without brain metastases. Genetic sequencing of cancer tissues in individual patients is useful in determining the risk of brain involvement. This may allow more efficient monitoring strategies to be developed.

**Abstract:**

Previous studies suggested that somatic *BRAF* and *NRAS* mutations in metastatic melanoma increase the risk for brain metastases. The risk related to other non-overlapping “driver” mutations is unknown. We performed a retrospective evaluation of the incidence, timing, and outcome of brain metastases in a population of melanoma patients that underwent uniform next-gen sequencing. All patients were treated with initial checkpoint inhibitor therapy. Seventeen of 88 patients (20.0%) developed brain metastases. Eleven patients had brain metastases at diagnosis (12.9%). These were all patients with *BRAF V600* or *NF1* mutations. Only six patients with *NRAS*, *NF1*, *KIT*, or *BRAF* mutations (including fusions/internal rearrangements experienced delayed CNS progression following immunotherapy (7.1%)). No “quadruple negative” patient developed brain metastases. Patients with brain metastases at diagnosis had a better outcome than those with delayed intracranial progression. Current predictive markers, (LDH, tumor mutation burden, and PDL1) were poorly correlated with the development of brain metastases. Treatment with immunotherapy appears to reduce the incidence of brain metastases. Next-gen molecular sequencing of tumors in metastatic melanoma patients was useful in identifying genetic subpopulations with an increased or reduced risk of brain metastases. This may allow eventual personalization of screening strategies.

## 1. Introduction

Patients with metastatic melanoma have a high risk of development of brain metastases [1]. In fact, the incidence proportion percentage of melanoma patients who develop brain metastases is currently higher in melanoma than in all other cancers [2]. Historically, melanoma brain metastases (MBM) were identified in about 28–33% of patients at the time of metastatic disease diagnosis [2,3]. An additional 40–44% of patients without initial brain metastases developed MBM within 4 years of starting initial treatment [4,5]. In the past, development of MBM was associated with increased morbidity and mortality [6,7]. 

Genetic testing of melanoma tumors has identified non-overlapping somatic mutations that active the MAP kinase pathway (*RAS*/*RAF*/*MEK*/*ERK*) in the majority of cutaneous melanoma patients [8,9]. The most common mutation occurs in *BRAF* gene at the V600 locus (usually at V600E). Other non-overlapping mutations have been described in *NRAS* (at the Q61 and G12/13 loci), as well as multiple different loss of function mutations in *NF1* [9,10]. *NF1* loss-of-function mutations increase *NRAS* signaling by reducing catabolism of RAS-GTP to RAS-GDP [11]. Rare MAP kinase pathway-activating mutations have also been identified in *CKIT* in cutaneous melanoma, although these are more commonly associated with mucosal and acral melanoma subtypes [12]. Mutations in *BRAF*, *NRAS*, *NF1*, and *CKIT* are sometimes referred to as “driver” mutations, as they rarely overlap each other in melanoma cells. Each mutation is independently capable of promoting growth, proliferation, and survival of melanoma cells [13]. Additional concurrent oncogene mutations are sometimes detected by next-gen sequencing. The effect of these “passenger” mutations on melanoma growth and progression is less well understood [13]. 

It has previously been reported that patients with *BRAF* and perhaps *NRAS* mutations may have an increased risk of central nervous system (CNS) involvement [14,15]. Relatively little is known about the contribution of other “driver” mutations identified via next-gen sequencing to the incidence, timing, or outcome of brain metastases in immune checkpoint inhibitor (ICI)-treated melanoma patients. Specifically, there is no current information concerning the risk of brain metastases in *NF1* or “quadruple negative” patients (no *BRAF*, *NRAS*, *NF1*, or *KIT* mutations). We therefore performed a retrospective review of our patient database to analyze the effect of melanoma genotype (defined by uniform next-gen sequencing) in a population of patients who received initial ICI therapy. 

## 2. Materials and Methods

### 2.1. Patient Identification

The identification of potential participants for this retrospective chart review was accomplished through a search of a Health Information Portability and Accessibility Act (HIPAA)-compliant iKnowMed medical record program (McKesson, Houston, TX, USA). We identified patients with metastatic melanoma who had received treatment with immune checkpoint inhibitors (ICI) by searching for patients treated with ipilimumab, nivolumab, or pembrolizumab.

Additionally, we obtained a separate list of our melanoma patients who had undergone next-gen sequencing for somatic tumor mutations using a single testing platform (Foundation Medicine CDx, Foundation Medicine, Cambridge, MA, USA). The patient lists were cross-referenced. ICI-treated patients harboring non-overlapping *BRAF*, *NRAS*, *NF1*, or *CKIT* mutations, as well as those who lacked these mutations (“quadruple negative”), were specifically targeted for further analysis. Patients with non-cutaneous melanoma sites (e.g., uveal, mucosal, and acral lentiginous melanoma) were excluded from this analysis. Patients who did not receive initial ICI treatment for metastatic disease were omitted. Patients without metastases, who received ICI treatment as adjuvant therapy following a complete surgical resection were also excluded.

### 2.2. Data Extraction

Eligible patient records were individually accessed, and patient data was extracted into a password-protected Excel spreadsheet (version 16.81, Microsoft, Redmond, WA, USA). Demographic data retrieved included an assigned unique patient number, age, and gender. Additional patient information extracted included driver mutation status (including *BRAF*, *NRAS*, *NF1*, *CKIT*-mutant or “quadruple negative”) and the presence of other concurrent somatic mutations within the tumor. When available, the tumor mutational burden (TMB), levels of PDL1 expression, and pretreatment lactate dehydrogenase (LDH) levels were recorded. Whether patients developed brain metastases at the time of diagnosis or developed delayed onset brain metastasis following ICI therapy was also noted. 

The type of ICI treatment the patient received was recorded. This information included the specific ICI regimen, treatment start date, cumulative number of doses, and treatment end date. Progression-free and overall survival was calculated from the treatment start date. If patients had not progressed, their data was censored at the date of the last clinic follow-up. All ICI-induced toxicities were documented. After the completion of data extraction from the patient care database into the study spreadsheet, patient identifying information was removed to preserve confidentiality. This study design was reviewed by the Western (WGC) Institutional Review Board (IRB) chair and was deemed exempt from a full IRB review.

### 2.3. Treatment Regimens

All patients with metastatic melanoma were treated initially with ICI, due to delays in obtaining next-gen sequencing data. The regimens utilized were based on the timing of regulatory approvals. Treatment consisted of standard doses of ipilimumab, pembrolizumab, or nivolumab as single agents, or the combination of ipilimumab plus nivolumab. We employed either the originally described regimen of ipilimumab plus nivolumab dosing [16], or an alternate (or “flipped”) dosing regimen [17]. ICI doses were generally rounded to the nearest higher vial size, due to the wide effective dose range and minimal differences in toxicity across a broad dose range for these agents. If patients achieved a radiologically or pathologically confirmed complete remission, elective treatment discontinuation was considered, as previously described [18]. 

If patients progressed following initial ICI therapy, clinical trial participation was suggested. If ineligible for clinical trials, patients with a *BRAF* mutation were treated with addition of a low-dose *BRAF* ± *MEK* inhibitor (typically consisting of dabrafenib at 75 mg/day with or without trametinib at 1 mg/day or alternatively encorafenib at 75 mg daily with or without binimetinib at 15 mg b.i.d.) with continuation of PD-1 antibody therapy, as previously described [19]. If patients had an *NRAS* or *NF1* mutation, the addition of a MEK inhibitor (trametinib, binimetinib, cobimetinib) with ongoing PD-1 therapy was offered [20]. 

### 2.4. Response Assessment

The best objective response (BORR) was assessed at 12 months from the start of therapy. Responses were described using RECIST 1.1 criteria [21]. A complete response (CR) required disappearance of all target and non-target lesions. Partial response (PR) was defined as more than a 30% reduction in the sum of bidimensional tumor measurements. Progressive disease (PD) was described as >20% increase in the sum of bidimensional tumor measurements or the development of new metastases. Stable disease (SD) was defined as any response not meeting criteria for CR, PR, or PD. Data collection concluded o 1 August 2022 (with a minimum potential follow-up of 18 months). 

### 2.5. Statistical Analysis

Descriptive statistics, such as median, standard deviation, and data range were calculated via the Excel spreadsheet. Overall survival (OS) and progression free survival (PFS) rates were analyzed using the Kaplan and Meier method [22]. Comparison between groups was performed using a log-rank test [23]. Analysis of potential predictive tests was performed using the Student’s *t*-test [24].

## 3. Results

### 3.1. Demographics

We identified 85 patients in our community oncology practice who underwent uniform next-gen tumor sequencing and were treated by a single physician (WS) with initial ICI therapy for metastatic cutaneous melanoma. Individual patient characteristics are described in Appendix A. The median age of patients in our series was 64.5 ± 14.6 years (±SD). The median duration of potential follow-up in this study was 36.9 ± 24.4 months. Our patients were treated with several different ICI regimens, related to the timing of regulatory approvals of these agents. Three patients received initial ipilimumab therapy, 33 received PD-1 directed monoclonal antibodies (7 pembrolizumab, 26 nivolumab monotherapy), and 49 received ipilimumab plus nivolumab (either the standard or “flipped-dose” regimen).

### 3.2. Mutation Frequency

Among our 85 patients, next-gen sequencing identified 5 patients (5.9%) with *BRAF* fusions or internal *BRAF* gene rearrangements. Twenty-eight (32.9%) had a *BRAF V600E/K/R* mutations, 23 (27.1%) had a variety of inactivating *NF1* mutations, and 15 (17.6%) had *NRAS* point mutations. As our patients were restricted to cutaneous melanoma, only one patient (1.2%) was found to have an activating *CKIT* mutation. A total of 13 patients (15.3%) without detectable *BRAF*, *NRAS*, *NF1*, or *CKIT* mutations were categorized as “quadruple negative”.

### 3.3. Time to Onset of Brain Metastases

Of our 85 patients who were treated with ICI, only 17 patients developed brain metastasis at any point during their therapy (20.0% overall incidence). The individual characteristics of the patients who developed brain metastases is shown (Table 1). These included nine men and eight women. The median age of patients with brain metastases was 59.0 ± 11.1 years. Of the 17 patients with CNS involvement in our series, 2 had *BRAF* fusions/rearrangements (40.0% of total *BRAF* fusion/rearrangement patients) and 9 patients had *BRAF V600* locus mutations (32.1% of all *BRAF* mutant patients). A total of four patients had an *NF1* mutation (17.4% of all *NF1* mutant patients), and one had an *NRAS* mutation (6.7% of all *NRAS* mutant patients). The only patient with a *CKIT* mutation developed delayed onset of brain metastases. No quadruple-negative patient developed brain metastases.

Time to diagnosis of brain metastases is shown (Figure 1). Eleven patients were diagnosed with brain metastasis as a component of their initial diagnosis of metastatic disease (12.9% of all ICI-treated patients), and only six patients developed delayed onset of brain metastasis following any form of ICI treatment (7.1% of all ICI-treated patients). All delayed-onset brain metastases occurred within 4 years from the start of ICI treatment.

### 3.4. Time to Onset of Brain Metastases by Genotype

The tumor mutation pattern appeared to play a significant role in the timing of onset of brain metastases. The incidence of brain metastases at diagnosis of metastatic disease or after initiation of ICI treatment is shown, based on “driver” mutation status (Figure 2). Patients with *BRAF* or *NF1* mutations were the only patients to present with brain metastases at the time metastatic disease was diagnosed. In patients with a *BRAF V600* mutation, eight/nine were found at the time of initial diagnosis of metastatic melanoma. Of patients with an *NF1* mutation three of four had brain metastases at diagnosis.

Delayed onset of brain metastases after the initiation of ICI treatment was a relatively rare event. This virtually always occurred in conjunction with systemic disease progression. Patients with *BRAF* fusions/rearrangement had a high rate of delayed brain metastases (40%), generally after progression following both ICI and TT treatment. Only 1 of 28 *BRAF V600* mutant patients developed delayed brain metastasis following the start of ICI treatment. One NF1 mutant patient developed delayed brain metastasis following ICI treatment. One patient with an *NRAS* mutation developed delayed onset of brain metastases following ICI therapy. The only patient with a *CKIT* mutation also developed delayed onset of brain metastases. None of the “quadruple negative” patients ever developed brain metastases. 

### 3.5. Brain Metastases and Survival

Individual treatment outcome data for the patients with brain metastases is shown (Table 2). A total of 17 patients either had brain metastases at diagnosis or developed delayed CNS progression. The median survival of all patients who developed brain metastases in this study was 14.1 months (range 1–52.2 + months). In contrast, median survival was 50.8 months in 68 patients who never developed brain metastases, *p* = 0.0002 (Figure 3A). 

### 3.6. Timing of Brain Metastases and Treatment Outcome

Of the 11 patients with brain metastases at onset, 4 of these patients are currently alive and disease free after elective treatment discontinuation. These four patients all had *BRAF V600* mutations. One additional *BRAF V600E* mutant patient achieved a mixed response, with isolated CNS progression. This patient had a biopsy-proven peripheral complete response, with development of one new brain lesion (which was recently ablated by cyberknife). This patient currently remains alive and remains on nivolumab maintenance therapy plus oral *BRAF* inhibitor. It should be noted that all brain metastases patients who achieved responses (*n* = 5) were initially treated with combined ipilimumab plus nivolumab therapy (median survival 10.5 months). At a median follow up of over 18 months, over 40% of the patients who had brain metastases at diagnosis are alive following ICI treatment. In contrast, all six patients who developed a delayed onset of brain metastases following ICI therapy have already died following disease progression (median survival 2.7 months, *p* = 0.0013) (Figure 3B).

### 3.7. Potential Biomarkers

Pretreatment lactate dehydrogenase (LDH), tumor mutation burden (TMB), and tumor cell PD-1 ligand (PDL1) expression are thought to correlate with immunotherapy responses and treatment outcome. An exploratory analysis of these potential biomarkers for the development of brain metastases was performed (Table 3). Pretreatment LDH, TMB, and PDL1 expression did not reach statistical significance as predictive biomarkers for development of brain metastases.

## 4. Discussion

Patients with metastatic melanoma historically have had a high incidence of brain metastases [1]. The percentage of metastatic melanoma patients who eventually develop brain metastases is highest in melanoma compared to other common cancers [2]. In the past, about 28–33% of metastatic melanoma patients were diagnosed with brain metastases at the time of diagnosis of metastatic disease [2,3]. An older trial randomized patients with metastatic melanoma (without brain metastases at enrollment) to either dacarbazine or temozolomide chemotherapy. In this trial, 20.6% of patients on temozolomide and 31.1% of dacarbazine-treated patients developed CNS progression within 1 year (not statistically different). By 3 years, the CNS failure rate approached 30–40% [5]. In a more recent publication evaluating metastatic melanoma patients treated between 2000 and 2012, the rate of CNS involvement was 31.7% prior to any systemic therapy [25]. An additional 35% of patients developed brain metastases during first-line treatment. There did not appear to be a significant difference in incidence of CNS relapse between patients receiving biochemotherapy, single-agent ipilimumab, anti-PD-1 or anti-PDL1 therapy, or *BRAF*-targeted therapy [25]. Overall survival in this study, however, appeared to be improved following either ICI or TT administration.

The process by which melanoma cells localize to the brain and form tumors is orchestrated by complex mechanisms [26]. These mechanisms appear to be regulated by both intrinsic genetic factors in tumor cells and microenvironmental influences (not further discussed herein). Several tumor factors are believed to be crucial, such as the existence of oncogenic *BRAF* or *NRAS* mutations [26]. For example, Colombino et al., identified frequent *BRAF* and *NRAS* mutations in tissue from MBM (in 48% and 23%, respectively) [27]. Fang et al. reported the mutation prevalence in biopsy tissue from MBM in 235 patients [28]. A total of 42% had *BRAFnon*-*V600K* melanoma, 14% had *BRAF V600K* tumors, 18% had *NRAS* mutations, and 26% were wild-type for both *BRAF* and *NRAS*. Unfortunately, the genetic testing methodology was not described in these reports.

Gino et al., reported the results of next-gen sequencing in 2067 melanoma biopsies from various sites of metastatic disease in the same patient. This included 132 brain metastases samples with 745 samples from matched primary tumors and 1190 matched non-CNS metastases [29]. This study identified *BRAF* (52.4%), *NRAS* (26.6%), *CDKN2A* (23.3%), *NF1* (18.9%), *TP53* (18%), *ARID2* (13.8%), *SETD2* (11.9%), and *PBRM1* (7.5%) as the most frequently mutated genes in brain metastases compared to other sites. The frequency of brain metastases related to the prevalence of each gene mutation was not determined.

It is not clear whether the somatic “driver” gene mutation pattern in metastatic melanoma cells influences the incidence of brain metastases. Some studies have suggested that the presence of a *BRAF V600E* mutation leads to an increased risk of brain metastases. For example, a multivariate analysis by Maxwell et al. found that *BRAF V600E* patients had a 2.24-fold increased risk of brain metastasis [14]. Jacobs et al., analyzed the *BRAF* and *NRAS* genotype by pyrosequencing of *BRAF* exon 15 (inclusive of codons 595 to 601) and *NRAS* exon 1 (codons 12 and 13) and exon 2 (codons 60 and 61) in 677 metastatic melanoma patients [30]. The frequency of brain metastases appeared higher in patients with *BRAF* mutations (24%) and *NRAS* mutations (23%), than in patients without these mutations (12%). In contrast, a retrospective review of 436 patients by Gummadi et al. found no difference in the incidence of brain metastases between patients with *BRAF*-mutated tumors versus those without a *BRAF* mutation (incidence rate ratio = 1.11) [31]. The incidence of brain metastases in NF1 mutant or “quadruple negative” patients is currently unknown.

In addition, little data is available concerning the timing of the onset of brain metastases related to melanoma genotype. Sperduto et al. evaluated the time from primary diagnosis to onset of brain metastases in patients with *BRAF*, *NRAS*, or *CKIT* mutations [15]. These investigators did not identify a difference in time to onset of brain metastases related to the tumor genotype. Survival after the diagnosis of brain metastases also appeared similar in these groups. An important caveat to this study is that the method for mutation testing was not described. Most patients also did not receive ipilimumab plus nivolumab therapy.

Based on our clinical experience, we hypothesized that the incidence of CNS progression in ICI-treated patients would be relatively low. We further hypothesized that patients who progressed in the brain after initial immunotherapy would have an adverse outlook compared to previously untreated patients with brain metastases, due to likely development of drug resistance. 

We therefore evaluated the frequency and timing of brain metastases in patients receiving initial ICI therapy. We determined that the overall rate of brain metastases in our ICI-treated patient population was surprisingly low (18.2%). This included 12.9% of patients with brain metastases at diagnosis of metastatic disease and only 7.1% that developed brain metastases following the start of ICI therapy. These results seem lower than previous estimates of the incidence of brain metastasis. 

A possible explanation is that this is due to the impact of earlier diagnosis and use of active ICI therapy for metastatic melanoma. This result may also be due to improvements in imaging technologies and increased use of CNS surveillance in metastatic melanoma patients. Our results support the conclusions of Franklin et al., who analyzed treatment results from 1704 patients. At 24 months follow-up, the incidence of brain metastases appeared to be reduced by first-line CKI therapy. The incidence of brain metastases was 30.3% with initial *BRAF* + *MEK* inhibitor therapy, 22.2% with CTLA4 + PD-1 treatment and 14.0% with PD-1 monotherapy. It should be noted that this data was from a large multi-institutional group, and genetic testing methods were not described.

In our study, patients with *BRAF* mutations had the highest incidence of brain metastases (43.2% if patients with *BRAF* fusions/internal rearrangements are included), followed by *NF1* mutant patients (16%). Based on very limited numbers of patients, the incidence of brain metastases in patients with *CKIT* mutations and *BRAF* fusions and internal rearrangements also appeared to be quite high. In contrast to previous reports [14,15], patients with *NRAS* mutations appeared to have a relatively low risk of brain metastases (11.8%). No patient with a “quadruple negative” genotype ever developed brain metastases. It should be noted that patients with other *BRAF V600* point mutations (e.g., three patients with *BRAF V600K* mutations) did not develop brain metastases in our 75-patient series. Since patients with these mutations may differ in clinical characteristics and treatment response [32], they will require evaluation in a much larger patient sample.

It should also be noted that in patients without MBM at diagnosis, there appeared to be a low risk of delayed brain metastases following ICI therapy. This risk was only 7.1% overall. This risk appeared to be reduced, regardless of whether patients received ICI monotherapy (with either CTLA4 or PD-1 antibodies) or combination therapy. This finding suggests that ICI treatment either reduces seeding of the brain from systemic sites or effectively controls microscopic metastases in the CNS. This intriguing observation requires further confirmation from prospective clinical trials.

The timing of the onset of brain metastases based on genotype is also not well characterized. Most patients either presented with brain metastases or developed them within the first 4 years after starting ICI therapy. Delayed CNS progression was generally associated with concomitant systemic disease progression. Isolated CNS progression was only seen in one ICI-treated patient. In addition, it is also notable that no ICI-treated patient developed meningeal carcinomatosis. 

Finally, our data indicate that a significant percentage of patients with CNS metastases at diagnosis were able to achieve durable responses and long-term survival. Most of these patients eventually underwent elective treatment discontinuation and remain in ongoing long-term remissions. In contrast, delayed onset of brain metastases universally occurred in the setting of systemic disease progression and appeared to indicate resistance to ICI therapy (and in most cases, TT resistance, as well). This form of CNS progression universally proved fatal. Novel treatment approaches need to be developed to more effectively treat patients with delayed CNS relapses after ICI therapy. 

Potential predictive factors for ICI response were evaluated to determine whether they were associated with an increased likelihood of brain metastases. Markers, such as LDH, TMB, and PDL1 TPS did not appear significantly different between patients who developed brain metastases and those who did not. It should be noted that these markers trended toward significance and this analysis may have been limited by the relatively small sample size. Further work to identify potential predictive markers for the development of brain metastases is clearly needed. 

The strengths of this study include use of a single next-gen tumor sequencing platform and treatment of all patients in a consistent manner by a single oncologist. The limitations of our study include a relatively small sample size (only 17 patients of our 85 patients developed brain metastases). The role of “passenger” mutations on the development and outcome of brain metastases will also require further evaluation in a larger patient sample. Thus, the influence of “passenger” gene mutations was not evaluated in the current study. It is also not clear whether the use of other next-gen sequencing platforms, including liquid biopsies, will lead to similar conclusions. We are currently performing a retrospective evaluation of brain metastases in a large multi-institutional cohort of metastatic melanoma patients to validate our results. The goal is to identify molecularly defined patient subsets with an elevated risk of brain metastases that require brain imaging at diagnosis. We hope to identify patient subsets that require more intensive CNS follow-up during the first 4 years after ICI treatment. We also are seeking to identify low-risk molecular subgroups that can safely undergo a de-escalation of CNS monitoring.

## 5. Conclusions

The frequency of MBM appears to be decreasing compared to previously published reports. In part, this may be due to earlier diagnosis and treatment of patients with ICI therapy and more frequent use of screening technologies. Next-gen sequencing of tumors in metastatic melanoma patients appears useful to allow identification of patients at elevated or decreased risk of brain metastases. Patients with *BRAF* and *NF1* mutations appear to have the highest risk of brain metastases at the time of diagnosis of metastatic disease. These patients should undergo CNS imaging during their initial pre-treatment evaluation. The rate of CNS progression after initial ICI treatment appears to be only 7% across all genetically defined subsets. Since this appeared to mainly occur at the time of systemic disease progression, it may be possible to decrease the frequency of CNS imaging in these patients. Patients with “quadruple negative” disease (no *BRAF*, *NRAS*, *NF1*, or *CKIT* mutations) appear to have a very low risk of brain metastases (either at diagnosis or following ICI therapy). These patients may not require CNS surveillance. In the future, a systematic next-gen oncogene testing approach may enable the personalization of CNS monitoring strategies in ICI-treated patients based on “driver” mutation genotyping.

## Figures and Tables

**Figure 1 cancers-16-00594-f001:**
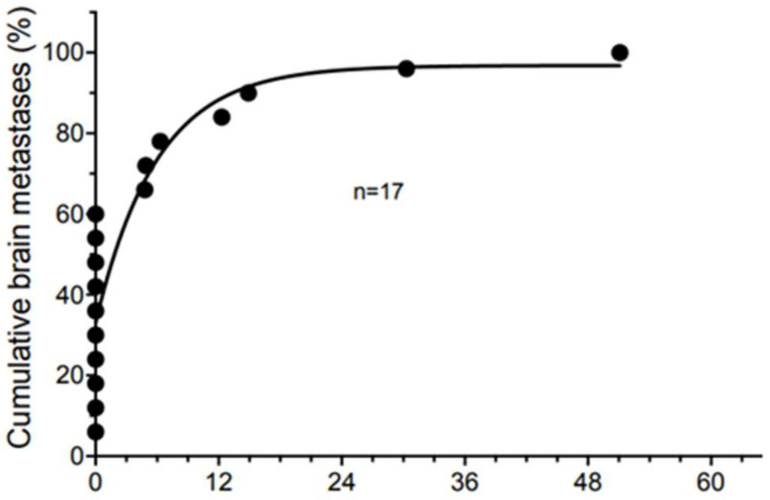
Time to diagnosis of brain metastases from onset of metastatic melanoma. Each point represents an individual patient.

**Figure 2 cancers-16-00594-f002:**
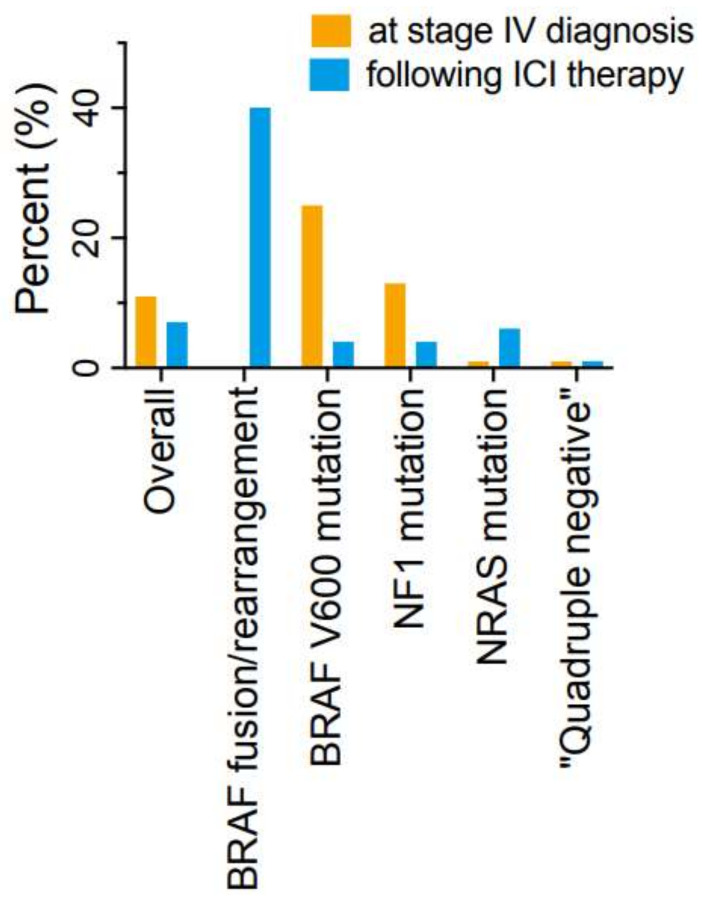
Incidence of brain metastases related to timing of diagnosis.

**Figure 3 cancers-16-00594-f003:**
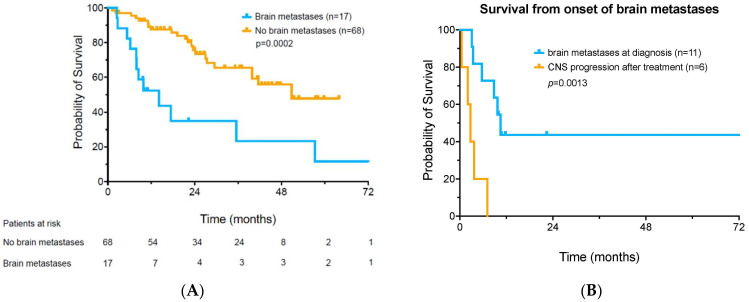
Survival of patients with metastatic melanoma. Panel (**A**): Survival of patients who developed brain metastases compared to those who did not. Panel (**B**): Survival of patients diagnosed with brain metastases at the time of onset of metastatic disease versus those who developed delayed onset of brain metastases after initial immunotherapy for metastatic melanoma. Survival was measured from the date of diagnosis of brain metastases.

**Table 1 cancers-16-00594-t001:** Patient demographics (patients with brain metastases).

UPN	Mutation	Age	Sex	Race	Primary Site	Site of Metastases	Stage	Initial LDH	PDL1 TPS %	TMB	Comorbidities
1	*BRAF T599_V600insT*	55	M	C	Extr	LN, lung	IVB	172	–	–	hypothyroidism, BPH
2	*BRAF NF2 fusion*	53	F	C	UNK	LN, adrenal	IVC	115	–	6	hypercholestrolemia
6	*BRAF V600E*	53	M	C	trunk	LN, sq, brain	IVD	286		8	HTN
7	*BRAF V600E*	45	M	C	UNK	brain	IVD	799	–	13	ulcerative colitis
8	*BRAF V600E*	59	F	C	Extr	brain, LN, abd	IVD	394	5	6	–
9	*BRAF V600E*	70	F	C	Trunk	brain	IVD	203	–	5	DM, HTN
10	*BRAF V600K*	52	F	C	Extr	LN, brain	IVD	164	1	19	osteoporosis, depression, hyperlipidemia
11	*BRAF V600K*	72	F	C	Trunk	LN, brain	IVD	226	60	11	HTN, Arthritis
12	*BRAF V600R*	62	F	C	UNK	LN, brain	IVD	276	1	11	HTN, depression, hyperlipidemia, hypothyroidism
13	*BRAF V600E*	34	M	C	neck	sq, liver, spleen lung, brain	IVD	251	–	24	–
14	*BRAF V600E*	43	F	C	trunk	adrenal, lung	IVC	223	0	6	depression
34	*CKIT*	72	M	C	trunk	sq, lung	IVB	163	2	0	SCCHN, superficial bladder cancer
35	*NF1*	77	M	C	scalp	lung, liver, bone, brain	IVD	463	4	–	DM, seizure
36	*NF1*	67	M	C	trunk	liver, brain	IVD	312	2	60	HTN, BCC
37	*NF1*	69	M	C	UNK	brain, lung sq	IVD	236	3	–	–
38	*NF1*	59	M	C	Extr	lung, bone	IVC	120	2	–	asthma
58	*NRAS*	63	F	C	trunk	LN, liver, lung	IVC	298	–	11	GERD

Abbreviations: UPN, unique patient number; LDH, lactate dehydrogenase; PDL1 TPS%, PD1 ligand tumor proportion score; TMB, tumor mutation burden per megabase; M, male; F, female; Caucasian; C, Extr, extremity; UNK, unknown primary; LN, lymph node; sq, subcutaneous; –, data not available; BPH, benign prostatic hypertrophy; HTN, hypertension; DM, diabetes mellitus; SCCHN, squamous cell carcinoma of the head and neck; BCC, basal cell carcinoma; GERD, gastroesophageal reflux disease.

**Table 2 cancers-16-00594-t002:** Treatment outcome (patients with brain metastases).

UPN	Timing of Brain Mets	ICI Regimen	ICI Doses	BORR	PFS (mo)	TT Added	OS (Months)	ICI Toxicity	Current Status
1	delayed	N	7	PD	4.3	–	14.1	nephritis	DOD
2	delayed	I	25	PD	6	D/T	57.3	hypothyroid	DOD
6	at onset	I + N	9	PD	2.5	E/B	9.8	–	DOD
7	at onset	I	8	CR	21.1	E/B	116	colitis	NED
8	at onset	I + N	6	PR	9.1	D/T	10	rash, colitis, encephalopathy	MR
9	at onset	I + N	11	CR	10.9	–	10.9	rash	NED
10	at onset	I + N	10	CR	9.6	–	22.3	hypothyroid	NED
11	at onset	I + N	6	CR	7.9	–	9.7	rash, colitis, hypopituitarism	NED
12	at onset	N	7	PD	7.1	–	7.8	pneumonitis	DOD
13	at onset	N	12	PD	27.6	E/B	35.5	–	DOD
14	delayed	I + N	7	PD	6.7	E/B	17.4	fever	DOD
34	delayed	I + N	17	PD	7.3	Nil	6.2	hot flash, colitis, fever, rash, hypothyroid	DOD
35	at onset	I + N	4	PD	2	–	2.5	diarrhea, rash	DOD
36	at onset	I + N	3	PD	1.8	T	2.7	hypothyroid	DOD
37	at onset	P	5	PD	5.7	–	7.9	uveitis	Died other
38	delayed	I + N	4	PD	3.2	–	5.3	colitis	DOD
58	delayed	I + N	7	PD	2.9	–	8.5	rash, hypophysitis, colitis	DOD

Abbreviations: UPN, unique patient number, ICI, immune checkpoint inhibitor, BORR, best objective response rate at 12 months; PFS, progression free survival; TT, targeted therapy; OS, overall survival; I, ipilimumab; N, nivolumab; P, pembrolizumab; PD, progressive disease; CR, complete response; D, dabrafenib; T, trametinib; E, encorafenib; B, binimetinib; Nil, nilotinib; DOD, died of disease; NED, no evidence of disease; MR, mixed response; Died other, died of non-cancer-related cause; –, data not available.

**Table 3 cancers-16-00594-t003:** Exploratory analysis of biomarkers for development of brain metastases.

	Brain Metastases			No Brain Metastases		
Marker	LDH (U/L)	TMB (per Mb)	Tumor PDL1 (%)	LDH (U/L)	TMB (per Mb)	Tumor PDL1 (%)
n	17	13	13	66	55	55
lowest	115.00	5.00	5.00	109.00	0.00	0.00
highest	799.00	82.00	82.00	1024.00	155.00	155.00
median	236.00	11.00	11.00	189.00	18.00	18.00
SD	162.70	25.81	25.81	119.16	38.96	38.96
*p* value *				0.17	0.12	0.12

LDH, lactate dehydrogenase; TMB, tumor mutation burden per megabase; PDL1, PD1 ligand. * by unpaired, two-tailed Student’s *t*-test comparing patients with and without brain metastases.

## Data Availability

Data are contained within the article and Appendix A.

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
