# Peer review of "The Effect of Non-Overlapping Somatic Mutations in BRAF, NRAS, NF1, or CKIT on the Incidence and Outcome of Brain Metastases during Immune Checkpoint Inhibitor Therapy of Metastatic Melanoma"

_cancers, 2024, doi:10.3390/cancers16030594_

Round 1
Reviewer 1 Report
Comments and Suggestions for Authors
The author presents an interesting retrospective study aimed at assessing the differences in somatic mutations in a cohort of ICI-treated metastatic melanoma patients. The paper is well written and clear. Some minor points to address:
1. Tx regimens: the author states "All patients with metastatic melanoma were treated initially with ICI, due to delays in obtaining Next-Gen sequencing data." Please clarify this part. In those patients with high tumor burden why wasn't targeted therapy with BRAF-MEKi used as front-line?
2. Statistical Analysis: the author states "Analysis of potential predictive tests was performed using student t- test". This sentence must be improved. T-test is a parametric test used to compare baseline differences between two groups. As survival analyses (both PFS and OS) were provided, it should be specified why a Cox regression was not used to assess the potential association of the clinical and histological features in study with survival. The sample size is large enough to perform a univariate Cox regression, which is the correct test to use when dealing survival data.
3. Did patients who received first line ipi+nivo have a higher disease burden at baseline? How was the choice of first-therapy made?
4. Survival graphs (fig. 3). Please add the number of patients at risk for each time point.
5. In the discussion the author covers the differences in terms of mutational profile in melanoma. However, the role of V600K is not mentioned, whilst it is known that such mutation is more frequent in elderly patients and in sun-damaged areas. Moreover its role in predicting the response to ICI has been object of recent studies (doi:10.3390/jcm11030828). Could the author comment on this aspect?
Author Response
We appreciate the thorough and thoughtful comments from the reviewers. We have tried to incorporate the reviewers’ comments to improve the manuscript.
Reviewer 1:
The author presents an interesting retrospective study aimed at assessing the differences in somatic mutations in a cohort of ICI-treated metastatic melanoma patients. The paper is well written and clear. Some minor points to address:
- Tx regimens: the author states "All patients with metastatic melanoma were treated initially with ICI, due to delays in obtaining Next-Gen sequencing data." Please clarify this part. In those patients with high tumor burden why wasn't targeted therapy with BRAF-MEKi used as front-line?
All patients received immunotherapy first, as results of Next-Gen oncogene sequencing was usually not available for 3-4 weeks and treatment was not delayed. More recently, this approach has been validated by the Alliance DREAMSEQ randomized trial. This included patients with bulky BRAF mutant melanoma, as these patients also can respond quite nicely to immunotherapy (especially combination ipilimumab plus nivolumab). It is not clear whether there is a survival advantage to initial cytoreduction with BRAF/MEK inhibitors in patients with bulky disease, followed by a switch to immunotherapy.
- Statistical Analysis: the author states "Analysis of potential predictive tests was performed using student t- test". This sentence must be improved. T-test is a parametric test used to compare baseline differences between two groups. As survival analyses (both PFS and OS) were provided, it should be specified why a Cox regression was not used to assess the potential association of the clinical and histological features in study with survival. The sample size is large enough to perform a univariate Cox regression, which is the correct test to use when dealing survival data.
This was an exploratory paired comparison of the numerical LDH, Tumor mutation burden and PDL1 score in tumor cells in patients with brain metastases versus those without using an unpaired two-tailed student t-test. We believe this to be an appropriate statistical test for comparison of numerical values, not related to patient survival, but to defined by the presence or absence of brain metastases. considering the relatively small patient and event (brain metastases) sample size. We recognize that ideally this should be correlated with survival in a Cox univariate or multivariate analysis. Unfortunately, this could not technically be performed on our patient sample due to the small number of patients with brain metastases. A Cox univariate and multivariate analysis is planned in a much larger data set from a cancer treatment group (currently underway).
- Did patients who received first line ipi+nivo have a higher disease burden at baseline? How was the choice of first-therapy made?
Patients were treated with best immunotherapy available based on the timing of regulatory approvals, thus earlier patients were treated with single agent PD-1 antibody, later patients received ipilimumab plus nivolumab following regulatory approval. Treatment was not selected based on tumor volume.
- Survival graphs (fig. 3). Please add the number of patients at risk for each time point. This was reasonable to add in Figure 3A comparing survival in patients with brain metastases compared to those who did not develop brain metastases. The numbers of patients in Figure 3B are too small after 1 year to be informative, and the graph is included only to make the point that a modest percentage of patients with brain metastases at onset have the potential for long term remission, while there were no long-term survivors among patients with delayed onset of brain metastases despite ongoing treatment.
Added to Figure 3
- In the discussion the author covers the differences in terms of mutational profile in melanoma. However, the role of V600K is not mentioned, whilst it is known that such mutation is more frequent in elderly patients and in sun-damaged areas. Moreover, its role in predicting the response to ICI has been object of recent studies (doi:10.3390/jcm11030828). Could the author comment on this aspect?
Added to text p9. Our small series only identified 3 patients with BRAF V600K mutations, none developed brain metastases, in contrast to BRAF fusions/internal rearrangements. Thus, the data set is too small to draw meaningful conclusions about patients with BRAF V600K patients.
Reviewer 2 Report
Comments and Suggestions for Authors
The article from Samlowski W. presents a retrospective analysis of the incidence, timing, and outcome of brain metastases in melanoma patients undergoing Next-Gen molecular sequencing and initial checkpoint inhibitor therapy.
Despite the clarity of the presentation, some critical considerations arise.
Firstly, the limited number of patients included in the study could pose a challenge to generalizing the obtained results. With only 88 patients considered, drawing definitive conclusions about the relationship between specific mutations (BRAF, NRAS, NF1, KIT) and the development of brain metastases becomes challenging. The small sample size may not be representative of the genetic diversity characterizing the population of patients with metastatic melanoma.
Furthermore, the lack of a significant correlation between current predictive markers (LDH, tumor mutation burden, and PDL1) and the risk of developing brain metastases raises doubts about the validity of these indicators in predicting this particular outcome. This aspect may prompt the need to explore additional, more specific predictive markers for brain involvement in the context of metastatic melanoma.
Despite the above-mentioned criticisms, the observation that immunotherapy appears to reduce the incidence of brain metastases is an intriguing result. However, a comprehensive understanding of the impact of this therapy would require further investigation on a larger and more diverse patient sample.
In conclusion, while the text provides valuable insights into the relationship between genetic mutations and brain metastases in metastatic melanoma, its practical utility could be enhanced with further research on a larger sample and a more thorough consideration of the validity of current predictive markers.
Author Response
We appreciate the thorough and thoughtful comments from the reviewers. We have tried to incorporate the reviewers’ comments to improve the manuscript.
Reviewer 2:
The article from Samlowski W. presents a retrospective analysis of the incidence, timing, and outcome of brain metastases in melanoma patients undergoing Next-Gen molecular sequencing and initial checkpoint inhibitor therapy.
Despite the clarity of the presentation, some critical considerations arise.
Firstly, the limited number of patients included in the study could pose a challenge to generalizing the obtained results. With only 88 patients considered, drawing definitive conclusions about the relationship between specific mutations (BRAF, NRAS, NF1, KIT) and the development of brain metastases becomes challenging. The small sample size may not be representative of the genetic diversity characterizing the population of patients with metastatic melanoma.
This limitation is discussed in text, we are following this up by analyzing a much larger multi-institutional 4200 immunotherapy treated metastatic melanoma patient cohort. Thus, the current analysis is intended to be hypothesis generating.
Furthermore, the lack of a significant correlation between current predictive markers (LDH, tumor mutation burden, and PDL1) and the risk of developing brain metastases raises doubts about the validity of these indicators in predicting this particular outcome. This aspect may prompt the need to explore additional, more specific predictive markers for brain involvement in the context of metastatic melanoma.
The prognostic markers we tested (Table 3) appear to correlate with the risk of progression and death in immunotherapy-treated melanoma patients (G. Long https://ascopubs.org/doi/10.1200/JCO.2021.39.15_suppl.9508). Our preliminary analysis suggests that these potential biomarkers are not so closely related to the risk of brain metastases. As this reviewer pointed out, this exploratory analysis needs to be followed up in a much larger cohort of patients. We are in the process of evaluating these correlates a much larger patient data set (~4200 patients).
Despite the above-mentioned criticisms, the observation that immunotherapy appears to reduce the incidence of brain metastases is an intriguing result. However, a comprehensive understanding of the impact of this therapy would require further investigation on a larger and more diverse patient sample.
We strongly agree and mentioned this in the discussion. We are currently analyzing a much larger multi-institutional patient data set. In addition, an analysis of the incidence of brain metastases as a site of failure in the DREAMSEQ trial randomizing between initial immunotherapy and BRAF/MEK inhibitor therapy in BRAF mutant melanoma is also being caried out, and seems to mirror our conclusions (personal communication, M. Atkins).
In conclusion, while the text provides valuable insights into the relationship between genetic mutations and brain metastases in metastatic melanoma, its practical utility could be enhanced with further research on a larger sample and a more thorough consideration of the validity of current predictive markers.
We thank the reviewer for their comments.